# Estimates of Household Food Waste by Categories and Their Determinants: Evidence from China

**DOI:** 10.3390/foods12040776

**Published:** 2023-02-10

**Authors:** Xiaolei Li, Yi Jiang, Ping Qing

**Affiliations:** 1College of Economics and Management, Huazhong Agricultural University, Wuhan 430070, China; 2The Research Center of Cultural and Tourism Industries, Wuhan Business University, Wuhan 430056, China

**Keywords:** household food waste, food categories, the incidence rate of food waste, the proportion of food waste, logit and Tobit model

## Abstract

Household food waste has become a global concern for all countries in the world. This study focuses on the household level to clarify the impact of household food waste, and we use an online questionnaire survey across China to estimate the proportion of household food waste from five categories, including whole food; fruits and vegetables; poultry, eggs, meat, and dairy products; staple food; and snacks and candy. Then, we use the logit and Tobit models to estimate the relationship between the five food categories and consumer characteristics. First, the statistical analysis shows that in China, the incidence rate and proportion of household food waste are 90.7% and 9.9%, respectively. Among them, the incidence rate and proportion of fruit and vegetable waste are the highest. The results of heterogeneity also show regional differences in the incidence rate and proportion of food waste. Second, the empirical results show that label cognition, garbage disposal cognition, vegetarianism, population, children or elders in the household, the experience of starvation, and age are important explanatory factors for the incidence and proportion of food waste in the household.

## 1. Introduction

According to the FAO report, global food waste amounts to 1.03 billion tons annually, accounting for about 17% of global food production. Food waste produce generates 3.30 billion tons of greenhouse gases annually, and the economic loss caused by food waste is about USD 750 billion annually [1]. In recent years, food waste has become a widespread concern in various countries facing severe challenges regarding food security, resources, the environment, and the economy. Nearly half of this food waste is generated at the household level [2]; therefore, analyzing food waste at the household level is an important guide toward addressing food waste at the consumption level.

A large body of literature on household food waste has analyzed waste measurement methods and drivers. The measurement methods of food include three types: the first method is the self-reported measurement, including a questionnaire survey [3,4], food waste diary [5,6], and interview [7]. The second method is to infer from the existing database [8,9]. The third method is the direct measurement of food waste, also known as a physical waste survey [10,11]. A large body of literature has adopted the self-report method to measure household food waste because of its timelessness and operationalization. Therefore, we also use a questionnaire survey to collect household food waste data of consumers nationwide in 2021, and to update information on food waste in Chinese households to a certain extent. In addition, the analysis of the drivers of household food waste also focuses mainly on the waste causes and demographic information. Studies have shown that food storage practices, food date label interpreting, and waste disposal cognition can intervene in consumers’ waste behavior [12,13,14]. Age, gender, income level, and education are also critical personal characteristics that concern the ways consumers produce food waste [15,16,17]. Our study considers all these aspects.

Reducing household food waste is a huge challenge, and the factors influencing food waste are complex. The generation of household food waste is related to household members [18], the amount and type of food brought into homes, and their shelf lives [9,12]. Most studies have taken a measure of household food without distinguishing the impact of food categories on consumer waste behavior. These kinds of literature make efforts to measure total household waste but have a limited contribution to reducing food waste. Studies have shown that the unique attributes of food categories will affect consumers’ choices [19,20]. Similarly, the unique attributes of these food categories will also affect consumers’ discarding behavior. For example, Williams et al. highlighted the different influences of food categories on food waste behavior [21]. Schmidt and Matthies indicated that the behavioral effects and performance levels varied across food groups [22]. As a result of categorizing foods, interventions can be targeted at the food item level. This has significant implications for the development of food waste reduction interventions.

To address the food waste issue, most studies on public strategies focus on information strategies that rely on consumer participation. However, macro strategies that consider only the entire consumer base do not consider taking consumer-level factors into account, implying that some existing messaging strategies must be more effective. Examples include China’s Clean Your Plate Campaign, the Food Waste Challenge in the United States, and New Zealand’s Love Food, Hate Waste Campaign. Based on this, our study focuses on food categories and adds value to the literature on estimating and addressing household food waste. This paper investigates consumers’ food waste behavior at the food category level and develops targeted food waste interventions. This paper is based on the largest food consumption market in China. We examine household food waste from the perspective of consumer consumption using the theoretical method of behavioral economics. Our study on consumers producing household food waste has implications globally, given the pervasive household food waste worldwide.

This study focuses on the household level: we adopt an online questionnaire survey to estimate the proportion of household food waste from five categories, including whole food; fruits and vegetables; poultry, eggs, meat, and dairy products; staple food; as well as snacks and candy. Then, we use the logit and Tobit models to estimate the relationship between the five categories of food and consumer characteristics and further clarify the impact of different categories in the household.

## 2. Literature Review

In the existing literature on household food waste, there are few studies on the waste of food categories, and most of them analyze the whole household food waste [8,9]. However, it is crucial to explore household food waste at the food categories’ level to provide more detailed suggestions for solving the food waste problem. Research has analyzed the influence of food categories on consumption choices [19,20,21]. The studies show that the food attribute differences in various aspects of food categories may change consumers’ choices. For example, Caldeira et al. found that grains, fruits, and vegetables were the most wasted food [23]. Ammann et al. and Ananda et al. also showed that fruits and vegetables had the most food waste [12,24]. At the same time, poultry, meat, eggs, and dairy products, as well as staple foods (such as rice and noodles), were the primary food categories with large consumption [25]. Packaged snacks were also the focus of attention in the food economy and nutrition health [26]. To sum up, our study’s food categories included not only whole foods; but also fruits and vegetables; poultry, meat, eggs, and dairy products; staple foods; as well as snack foods.

The driving factors influencing household food waste are complex and varied. Our online questionnaire collected many data related to food behavior and cognition to clarify the drivers that influence household food waste. We also combined demographic information to explore how these factors affected various food categories of waste.

Food storage knowledge and practice are critical to reducing household food waste [12,13,22]. The optional storage conditions may vary according to the characteristics of different food categories [6]. Brown et al. considered that reducing the refrigerator temperature can extend the shelf life of food [27]. Ananda et al. also showed that consumers with more knowledge of food storage would produce less food waste [12]. However, Kavanaugh and Quinlan showed that about four out of five consumers claimed that they “often or always” check the food date label, and consumers’ misinterpretation of food date labels can lead to the waste of food that is safe to eat [28]. Turvey et al. found that only about half of the consumers correctly understood the definition of the food date label when purchasing food [29]. FMI found that about 37% and 25% of consumers misinterpret food date labels, leading to discarding unspoiled food by the day on the use-by and sell-by labels, respectively [30]. Food waste caused by misinterpreting food date labels accounts for about 33% of total food waste [31]. Consumers’ cognition of food waste disposal knowledge will affect their decision-making behavior. For example, landfill disposal information will reduce food waste, while composting disposal information will increase food waste [32].

In the literature on consumer characteristic factors, some studies have found that consumers with vegetarianism motivations can influence their household food choices and diet [33]. Income is an important determinant of household food waste, and increasing income leads consumers to produce more food waste [34]. High-income groups and developed countries are more prone to food waste at the consumption stage because consumers tend to throw away more edible food [35]. According to the study of Li et al. and Jiang et al. household size is one of the critical determinants of food waste in Chinese rural households [36,37]. The number of household members is directly proportional to food waste [38]. Families with children produce higher total food waste at the household level [39]. Generally, individuals over 65 tend to produce less food waste [18]. In addition, due to the long-term urban–rural binary division, Chinese households’ cultural background and environment also show significant differences between urban and rural areas, which may also affect household food waste [37]. According to Qi et al. rural households waste significantly more food than urban households, while the gap between urban and rural food waste is reducing [40].

Chen and Zhou analyzed the impact of the starvation experience on food consumption and nutritional health [41]. Studies on the impact of gender on food waste have also been inconsistent. Some studies indicated that females produced less food waste [18], gender had no significant effect, or females produced more food waste [42]. Studies have shown that household food waste is negatively associated with age [43,44]. The education level of consumers is directly proportional to food waste [45,46]. These studies suggested that the higher the education level of the leading family members (especially the head of the household), the higher the overall income level of the household, and thus the more food purchases and food waste are likely. Studies have found that household food waste varies significantly due to different cultural backgrounds. For example, there are differences in household food waste across regions [47]. Our study also considered the influence of these consumer characteristics.

To sum up, this paper studied the impact of household waste in China. In this process, we estimated the proportion of whole food and four food categories, and we also considered the whole food and four food types, which may produce behavioral differences under the impact of consumers’ characteristic factors.

## 3. Questionnaire Design

### 3.1. Data Collection

Our research relied on data collected from an online survey targeting general food consumers in China. Specifically, fruits and vegetables, poultry, meat, eggs, and dairy products, as well as staple foods, are the primary food categories with large consumption in China [8]. The first part of the survey included three fill-in-the-blank and single-choice questions designed to investigate food shopping habits and food perceptions. The second part contained five fill-in-the-blank questions designed to investigate the proportion of household food waste in the previous week. The last part of the survey included consumer understanding and perception of food waste treatment, consumer demographics, and socio-economic information. These ten questions were in the form of fill-in-the-blank and single choice (See Appendix A). Following Malong and Lusk, we set two trap questions in the questionnaire to reduce the possibility of nonattention [48]. Consumers who did not pass the trap questions were excluded from the final data set.

In the survey, the second part of the questionnaire questions took the following form: “In general, how much of the following food categories has been discarded in your household in the previous week (the waste scenario below does not consider passive discarding due to packaging damage or contamination)”.

In the survey design stage, we performed six focus group interviews in the survey design stage, each containing five to seven individuals. Members invited to the interviews included food science experts, food enterprise managers, professionals, food market regulators, and general consumers. The focus groups helped us finalize the food categories and measurement methods. Before launching the final survey, we conducted five rounds of pilot surveys, which allowed us to improve the language clarity and flow of the survey. The final survey was conducted online through Lediaocha, a sizeable professional market research firm in China. Formal sample collection began at the end of September 2021 and was completed by the end of October 2021. The samples were collected across China based on the population proportion in each Chinese province/provincial level metropolitan or autonomous region according to China’s Seventh Population Census. A total of 1829 respondents returned the survey. After removing respondents who did not answer any of the choice experiment questions or did not pass the attention check, our final sample contained 1746 valid responses.

### 3.2. Food Categories

For the analysis, the survey collected self-reported food waste data in five categories: (1) whole foods; (2) fruits and vegetables (raw and cooked); (3) poultry, meat, eggs, and dairy products (raw and cooked); (4) staple food such as rice and noodles (raw and cooked); (5) snacks and candy (not including the above foods).

## 4. Research Methodology

### 4.1. Logit Model

The logit model is one of the models of the discrete choice method, which belongs to the category of multivariate analysis. Since the dependent variables are discrete, they contain only 1 and 0. Therefore, it is also called the binary choice model. The formula of the cumulative distribution function of the logical is as follows:(1)$$P\left(y=1|x\right)=F\left(x,\beta \right)=\wedge \left(x^{\prime }\beta \right)=\frac{\exp\left(x^{\prime }\beta \right)}{1+\exp\left(x^{\prime }\beta \right)}$$

The specific formula of the logit model is as follows:(2)$$\pi_{i}=\beta_{1}^{\prime }\gamma +\beta_{2}^{\prime }\delta +\varepsilon_{i}$$
where the dependent variable $\pi_{i}$ is a binary variable, and =1 indicates that the percentage of food waste generated by the respondent’s household in the previous week is not equal to 0, otherwise = 0. Furthermore, $\beta_{1}^{\prime }$ and $\beta_{2}^{\prime }$ represent the estimated coefficient vector; γ represents the main independent variables, including store knowledge, label knowledge, and process knowledge; δ represents the consumer characteristics’ variables, including vegetarian, income, population, child and elder, rural, starve, female, age, and education. $\varepsilon_{i}$ is a random perturbation that follows a normal distribution.

### 4.2. Tobit Model

We assume that the individual is i, i=1,…, N. $y_{i}$ represents the percentage of food waste from the various categories to the total food purchased by the household in the previous week. Since the percentage of food waste is non-negative, we further assume that the latent variable of the percentage of food waste, $\;y_{i}^{*}$, is left-censored at zero. The dependent variable $y_{i}$ and the underlying variable $y_{i}^{*}$ have the following relationship by Equation (3):(3)$$y_{i}=\left\{\begin{array}{c} y_{i}^{*},\;y_{i}^{*}>0\\ 0,\;y_{i}^{*}\leq 0 \end{array}\right.$$

The latent variable $y_{i}^{*}$ satisfies Equation (4):(4)$$y_{i}^{*}=\partial^{\prime }X+\varepsilon_{i}$$
where vector $X=\left(x_{1},\ldots x_{n}\ldots ,x_{N}\right)$ contains all dependent variables that could affect household food waste. ∂ represents a vector of coefficients to be estimated. $\varepsilon_{i}$ is a random disturbance that follows a normal distribution.

The log-likelihood lnL is given as:(5)$$lnL=\overset{I}{\underset{i=1}{\sum }}\left\{\left(1-D_{i}\right)\ln\left[1-\varphi \left(\frac{\partial^{\prime }X}{\sigma }\right)\right]+D_{i}\ln\left[\frac{1}{\sigma }\emptyset \left(\frac{y_{i}-\partial^{\prime }X}{\sigma }\right)\right]\right\}$$
where $D_{i}=\left\{\begin{array}{c} 1,\;y_{i}>0\\ 0,\;y_{i}=0 \end{array}\right.,\;\varphi \left(\cdot \right)$ and ∅(⋅) represent the standard normal distribution function and the probability density function, respectively.

To explore the impact of consumer characteristic variables on the percentage of household food waste, we decompose $\partial^{\prime }X$:(6)$$y_{i}^{*}=\beta_{1}^{\prime }\gamma +\beta_{2}^{\prime }\delta +\tau_{i}$$

The structure of independent variables is the same as that of the above logit model.

### 4.3. Summary Statistics

Table 1 reports the descriptive statistics. The sample means the value of dependent variables also describes the incidence rate and proportion of food waste in each of the five categories. Among the independent variables, the sample mean of *Store_cognitive* was 0.8, which indicates most consumers could store foods properly, and the sample mean of *Label_cognitive* was 0.1, which shows that only a small number of consumers correctly interpreted the definition of the food date label. In addition, the sample mean of *Treat_cognitive* was 0.5, indicating that respondents were neutral in their view that waste disposal can reduce wasteful behavior. In the table, we also report the mean and standard deviation of variables related to consumer characteristics. Furthermore, 8.6% of households had *vegetarians.* The average annual household *income* was CNY 201,400, and the average household had about 3.4 people. The sample mean of *child or elder* and *starve* were 0.8 and 0.1, respectively. Approximately 7.6% of respondents lived in urban areas, and 54.8% of our respondents were *female.* The average respondent’s *age* was 33.6. We had disproportionally more respondents with a college *education*. Both education and income were higher than the national average.

## 5. Results

### 5.1. Estimation of Household Food Waste

Table 2 compares the incidence rate of household food waste in five categories across regions. We looked at the incidence rate of household food waste in the overall sample, six administrative regions, and urban and rural areas. The incidence rate of household food waste was calculated by dividing the food waste sample size by the total sample size. Regarding the overall waste incidence rate, food waste occurred in most households, with an incidence rate of approximately 90.7%. Fruits and vegetables had the highest incidence rate of food waste, at around 87.7%. Following that was the rate of food waste in poultry, meat, eggs, and dairy products, as well as staple foods, at about 69.5% and 61.9%, respectively. In addition, snacks had the lowest incidence rate of food waste, at around 59.9%.

Regarding the incidence rates of household food waste in the six administration regions, the northeast region had the highest rate of whole food waste, at approximately 94.1%. Fruits and vegetables, poultry, meat, eggs, and dairy products had the highest waste incidence rates in north China, at around 87.7% and 72.2%, respectively. Fruits and vegetables, staple foods, and snack foods had the highest waste incidence rates in the central south, at approximately 87.7%, 66.2%, and 65.35%, respectively. In terms of the incidence rate of food waste in urban and rural areas, rural areas had a higher incidence rate of the five categories than urban areas.

Table 3 compares household food waste proportions in the five categories across regions. We examined the household food waste proportions in the overall sample, six administrative regions, and urban and rural areas. Food waste proportion refers to the percentage of food waste estimated by consumers in the past week. In terms of food waste percentage in the overall sample, the total household food waste percentage was 9.9%. That corresponds to the 10% of food wasted per household reported by CCTV news [49]. Among the food categories, first, fruits and vegetables accounted for 13.2% of household food waste, and this high proportion of fruit and vegetable waste is directly related to their perishability. The higher percentage of food waste from fruits and vegetables is consistent with the findings of some household food waste studies. Ananda et al. showed that fruits and vegetables accounted for approximately 48% of food waste in terms of weight [12]. In the food waste study in Switzerland, Ammann et al. reported that fruits and vegetables and peels’ category received the largest food waste [24]. According to Williams et al. the category with the largest amount of waste (30 %) was fruits and vegetables [21]. Second, poultry, meat, eggs, and dairy products accounted for 8.4% of household food waste. The high waste rate indicates that this category of food is consumed in greater quantities by consumers in their daily lives. Third, the percentage of staple foods wasted in household was 6.9%. The low proportion of staple food waste indicates that the food can be reused to reduce the possibility of waste. Similarly, Ananda et al. showed that bakeries represented the lowest food waste in Australia in terms of weight [12]. Fourth, the percentage of household food waste from snacks was 7.5%. It is important to note that most snack foods are pre-packaged and have a long shelf life. Consumers’ misinterpretation of the date labels may account for 7.5% of the waste, resulting in inappropriate discarding behavior.

In terms of the proportion of food waste in the six administration regions, the southwest administration region had the highest proportion of whole food waste at 11.6%. Among the food categories, the central south region had the highest proportion of fruit and vegetable waste, at about 14.4%. The southwest region had the highest proportions of poultry, meat, eggs, and dairy products, as well as staple food waste, which were about 9.9% and 8.5%, respectively. The east China region had the highest proportion of snack food waste, at about 8.8%.

### 5.2. Determinants of Household Food Waste

Table 4 reports the logit model results for the incidence rate of household food waste. We used data from five food-type models, including whole foods (whole); fruits and vegetables (fruit); poultry, meat, eggs, and dairy products (meat); staple foods (staple); as well as snacks and candy (snack). The results showed that coefficients of *Store_cognitive* and *Label_cognitive* in all models were insignificant. Furthermore, the coefficients of *Treat_cognitive* were positively significant in the meat, staple, and snack models. We also found that the coefficients of *vegetarian* were positively significant in the meat, staple, and snack models, indicating that vegetarians do not eat these three categories of food, resulting in a significant amount of household waste. We considered that households with *vegetarians* were more likely to waste on poultry, meat, eggs, and dairy products, staple foods, as well as snacks and candy than households without *vegetarians*. The coefficients of *population* were only positively significant in the fruit and staple models; this is consistent with the previous study findings [38]. In addition, except for the fruit model, the coefficient of *child or elder* was significant in the other models. Given that the majority of Chinese households consist of three generations of children, parents, and elderly people living together, having children and elderly people in the household increases the number of people in the household while also resulting in more food waste. This is not the case in most other countries where the elderly live alone [18,39]. The coefficients of *starve* and *age* were significant in all models. It is worth noting that consumers who experience hunger tend to waste food. We believe that previous hunger experiences may increase food stocking by consumers, with large amounts of fresh food going uneaten for a period of time and spoiling, followed by food waste [34,43]. However, the *income, rural, female*, and *education* coefficients were insignificant in all models.

Table 5 reports the Tobit model results for the proportion of household food waste. The results showed that the coefficients of *Store_cognitive* were insignificant in all models. Our findings differ from those of Hoek et al. and Schanes et al. who concluded that knowledge of food storage is important for reducing household food waste [13,14]. The reason for this could be that respondents have more knowledge about food storage (as shown in Table 1). *Label_cognitive* was positively significant in the whole food waste model at 1% levels and marginally significant for the meat model, but not in the other three models. This suggests that misinterpretation of food date labels can contribute to household food waste [28], and given the differences in food category characteristics, the impact of food date label perceptions on food waste varies by food category [31]. *Treat_cognitive* was associated with fruit and vegetable waste and was insignificant in the other four models. On the one hand, fruits and vegetables have a high perishability and are prone to waste [12]. On the other hand, respondents believed that food waste disposal strategies could be a good solution for food waste generated in the household, increasing the likelihood that they would discard that food category [32]. The coefficient of *vegetarian* was positively significant in all models at 1% levels. *Child or elder* had a significant and positive connection with whole food, staple, and snack waste. Having experience with starvation (*starve*) positively correlated with five types of food waste. Moreover, with an increase in *age*, the proportion of food waste in the five categories will decrease. In addition, we found that *income, population, rural, female,* and *education* were irrelevant to the proportion of food waste in the five categories.

## 6. Conclusions and Implication

### 6.1. Conclusions

Using data from China, our findings show that more than 90% of households waste food, with fruits and vegetables having the highest rate of food waste in China. The average amount of food wasted per household is 9.9%. This corresponds to the 10% of food wasted per household reported by CCTV news [49]. Fruits and vegetables account for the greatest proportion of food waste in households where food waste occurs. However, the proportion of food waste in snacks is higher than in staple foods, which may be due to consumers misinterpreting food date labels, resulting in inappropriate food-discarding behavior and waste. Food waste trends across food categories are consistent with international experience [1,6]. The results of heterogeneity show that the incidence and proportion of food waste in the five food categories differ significantly across the six administrative regions, with rural areas having a higher incidence and proportion of food waste than urban areas.

Second, the empirical results of food waste-influencing factors show that label cognition, garbage disposal cognition, vegetarianism, population, children or elders in the household, the experience of starvation, and age are important explanatory factors for the incidence and proportion of food waste in the household. Specifically, among the main independent variables, knowledge of compost information (waste disposal perception) is associated with increased household food waste incidence, food date label misinterpretation (label perception), and knowledge of compost information (waste disposal perception) with an increased household food waste proportion. Vegetarians, the population, children or elders, the experience of starvation, and age are all associated with the incidence and proportion of food waste.

### 6.2. Implication

Food waste is a global issue, and an effective measure to reduce food waste is to identify the food categories and regions with the high levels of food waste and focus on solving that type of food waste. First, we consider storage interventions as a strategy for reducing fresh food waste. On the one hand, interventions should extend food shelf life and food spoilage prevention campaigns, i.e., the sealing of plastic packaging for fresh vegetables, fruits, and meat. On the other hand, airtight packing should also consider using resealable, easily emptied packing and more-capacity packaging sizes. Financial incentives could be offered to manufacturers to encourage them to optimize their food packaging design and thus reduce food waste. Second, we consider that interventions that provide consumers with information and guidance on food processing are a strategy that could reduce the waste of cooked foods. For example, a course APP that provides food cooking techniques and portions can reduce food waste caused by poor or excessive cooking quality. Third, more information strategies to reduce food waste in rural areas should be established, such as the Clear Your Plate Campaign, which aims to raise consumer awareness about food waste in rural areas.

According to the empirical results, consumers should first be educated on the food date label in order to correct consumers’ misinterpretation of the label definition, and furthermore, avoid the resulting appropriate discarding behavior. Second, the outer packaging label of pre-packaged food should be improved. Consumers could understand the date label more directly if the definition of the date label was marked on the outer packaging. Moreover, new labeling technologies should be used. Intelligent labels, for example, can show the current quality status of food and the conditions under which it can no longer be consumed. Third, we should consider reducing the amount of information available about composting food waste treatment. Fourth, when formulating strategies to address food waste, demographic information should be taken into account. Food waste strategies must be precisely targeted and implemented by the population.

Our study has some limitations. First, we used self-reported food waste data for the analysis, which may lead to consumers underestimating actual household food waste. We avoided this bias as much as possible by asking consumers about their household food waste over the past week. Second, our categories of food waste are limited, and this paper found evidence that there are statistically significant differences among food categories concerning household food waste.

## Figures and Tables

**Table 1 foods-12-00776-t001:** Descriptive statistics.

Variables	Descriptive	Mean	S.D.
Dependent Variables			
Incidence_whole	=1 indicates that the percentage of the whole food waste generated by the respondent’s household in the previous week is not equal to 0, otherwise = 0	0.907	0.291
Incidence_fruit	=1 indicates that the percentage of fruit and vegetable waste generated by the respondent’s household in the previous week is not equal to 0, otherwise = 0	0.877	0.328
Incidence_meat	=1 indicates that the percentage of produced poultry, meat, eggs, and dairy products waste generated by the respondent’s household in the previous week is not equal to 0, otherwise = 0	0.695	0.461
Incidence_staple	=1 indicates that the percentage of staple food (such as rice and noodles) waste generated by the respondent’s household in the previous week is not equal to 0, otherwise = 0	0.619	0.486
Incidence_snack	=1 indicates that the percentage of snack and candy waste generated by the respondent’s household in the previous week is not equal to 0, otherwise = 0	0.599	0.490
Waste_whole	A continuous variable representing the percentage of the whole food waste to a household’s total food purchase in the previous week (unit: %)	9.895	9.109
Waste_fruit	A continuous variable representing the percentage of fruit and vegetable waste to a household’s total food purchase in the previous week (unit: %)	13.185	14.425
Waste_meat	A continuous variable representing the percentage of poultry, meat, eggs, and dairy product waste to a household’s total food purchase in the previous week (unit: %)	8.405	10.928
Waste_staple	A continuous variable representing the percentage of staple food (such as rice and noodles) waste to a household’s total food purchase in the previous week (unit: %)	6.850	9.465
Waste_snack	A continuous variable representing the percentage of snack and candy waste to a household’s total food purchase in the previous week (unit: %)	7.505	11.152
Main independent variables			
Store_cognitive	=1 if the respondents are correct about milk, pork, toast, eggs, and apple storage, otherwise = 0	0.840	0.367
Label_ cognitive	=1 if the respondents correctly interpret the definition of food date label, otherwise = 0	0.062	0.242
Treat_ cognitive	=1 if the respondents believe food waste disposal strategies can reduce food waste behavior, otherwise = 0	0.531	0.499
Consumer characteristic variables			
Vegetarian	=1 if there are vegetarians in the household, otherwise = 0	0.086	0.280
Income	A continuous variable representing respondents’ household annual pre-tax income, per 1000	201.420	12.868
Population	A continuous variable representing respondents’ household size	3.379	1.252
Child or elder	=1 if the respondents’ household has child or elder, otherwise = 0. Children are required to be under 18-years-old, and elders are required to be over 60-years-old	0.763	0.425
Rural	= if the respondents’ household lives in an urban area, otherwise = 0	0.763	0.425
Starve	=1 if the respondents have experienced passive starvation for more than one month, otherwise = 0	0.126	0.332
Female	=1 if the respondent is a female. Otherwise = 0	0.548	0.498
Age	A continuous variable representing respondents’ age	33.614	8.962
Education	A cardinal variable measured with seven levels: 1 for incomplete primary school, 2 for primary school graduate, 3 for middle high school graduate, 4 for high school (technical school/vocational high school) graduate, 5 for 2-year college graduate, 6 for 4-year university graduate, and 7 for Masters or above graduate	5.684	0.847

**Table 2 foods-12-00776-t002:** The table compares the incidence rate of household food waste across regions (unit: %).

Regions	Whole	Fruit	Meat	Staple	Snack
Overall	90.7	87.7	69.5	61.9	59.9
North China	90.3	87.7	72.2	56.7	59.0
Northeast	94.1	86.1	66.3	58.7	55.3
East China	90.3	86.8	69.6	60.3	61.9
Central South	90.3	87.7	71.5	66.2	65.3
Southwest	84.8	86.2	68.5	61.4	58.6
Northwest	87.6	86.6	61.2	52.7	54.0
Urban	90.7	87.9	67.6	59.6	59.0
Rural	94.6	94.6	75.7	73.0	64.9

**Table 3 foods-12-00776-t003:** The table compares household food waste proportions across regions (unit: %).

Regions	Whole	Fruit	Meat	Staple	Snack
Overall	9.9	13.2	8.4	6.9	7.5
North China	10.5	13.7	7.8	6.6	7.0
Northeast	10.8	12.6	8.0	6.5	7.2
East China	9.9	13.0	9.0	6.9	8.8
Central South	10.8	14.4	9.4	8.0	8.5
Southwest	11.6	14.2	9.9	8.5	8.5
Northwest	8.6	11.7	8.5	6.2	7.1
Urban	9.9	13.6	8.6	6.7	7.2
Rural	11.0	13.6	10.4	8.7	8.0

Note: Food waste proportion refers to the percentage of food waste in the past week estimated by consumers. The results are the mean value of the estimates.

**Table 4 foods-12-00776-t004:** Logit result for incidence rate of household food waste.

Variables	Whole	Fruit	Meat	Staple	Snack
Store_cognitive	0.022	−0.290	−0.115	−0.013	−0.110
	(0.235)	(0.226)	(0.152)	(0.143)	(0.143)
Label_cognitive	0.596	0.298	0.440 *	0.282	0.255
	(0.415)	(0.340)	(0.244)	(0.217)	(0.219)
Treat_cognitive	0.257	0.110	0.353 ***	0.267 ***	0.287 ***
	(0.170)	(0.149)	(0.108)	(0.103)	(0.102)
Vegetarian	0.459	0.755 *	1.128 ***	1.141 ***	1.177 ***
	(0.413)	(0.397)	(0.268)	(0.239)	(0.239)
Income	0.000	0.005	0.002	0.001	0.004
	(0.007)	(0.006)	(0.004)	(0.004)	(0.004)
Population	0.132	0.240 **	0.017	0.141 **	0.028
	(0.104)	(0.098)	(0.071)	(0.058)	(0.065)
Child or elder	0.546 ***	0.159	0.347 **	0.237 *	0.466 ***
	(0.211)	(0.196)	(0.151)	(0.137)	(0.143)
Rural	0.083	0.106	−0.104	−0.157	−0.037
	(0.273)	(0.243)	(0.164)	(0.155)	(0.155)
Starve	1.033 ***	1.195 ***	0.588 ***	0.786 ***	0.791 ***
	(0.382)	(0.350)	(0.192)	(0.183)	(0.178)
Female	−0.184	0.149	0.023	−0.092	−0.082
	(0.172)	(0.152)	(0.109)	(0.104)	(0.103)
Age	−0.037 ***	−0.032 ***	−0.033 ***	−0.028 ***	−0.031 ***
	(0.011)	(0.009)	(0.007)	(0.007)	(0.007)
Education	−0.119	−0.134	−0.000	−0.017	−0.064
	(0.117)	(0.107)	(0.077)	(0.076)	(0.073)
Constant	3.200 ***	2.798 ***	1.355 **	0.620	1.096 *
	(1.026)	(0.943)	(0.679)	(0.648)	(0.640)
AIC	1062.657	1265.953	2075.087	2233.670	2258.630
Log-Likelihood	−518.329	−619.976	−1024.544	−1103.835	−1116.315
Observations	1746	1746	1746	1746	1746

Note: robust standard errors in parentheses; *, **, and *** indicate significance at the 10%, 5%, and 1% significance levels, respectively.

**Table 5 foods-12-00776-t005:** Tobit result for the proportion of household food waste.

Variables	Whole	Fruit	Meat	Staple	Snack
Store_cognitive	−0.830	−0.460	−1.880 *	−0.482	−1.189
	(0.712)	(1.056)	(1.118)	(1.056)	(1.372)
Label_cognitive	4.609 ***	0.167	2.665 *	0.977	1.055
	(1.211)	(1.257)	(1.362)	(1.458)	(1.504)
Treat_cognitive	0.380	−1.329 **	1.475 *	1.120	1.391 *
	(0.410)	(0.666)	(0.797)	(0.692)	(0.829)
Vegetarian	3.202 ***	3.765 **	6.218 ***	6.058 ***	6.967 ***
	(0.920)	(1.581)	(1.348)	(1.090)	(1.192)
Income	0.015	0.022	0.054 *	0.047 *	0.035
	(0.019)	(0.030)	(0.030)	(0.028)	(0.043)
Population	−0.107	0.103	0.380	0.526 *	0.477
	(0.172)	(0.267)	(0.428)	(0.295)	(0.405)
Child or elder	2.234 ***	−0.156	1.877 *	2.140 **	2.667 **
	(0.653)	(1.190)	(1.114)	(0.909)	(1.172)
Rural	0.369	1.442	1.059	−0.333	−0.880
	(0.842)	(1.581)	(1.199)	(1.161)	(1.502)
Starve	3.007 ***	1.903 **	4.074 ***	3.993 ***	4.408 ***
	(0.736)	(0.891)	(1.003)	(0.865)	(1.359)
Female	−0.203	1.364 *	−0.736	−0.994	0.119
	(0.518)	(0.725)	(0.600)	(0.642)	(0.733)
Age	−0.106 ***	−0.076	−0.185 ***	−0.175 ***	−0.223 ***
	(0.033)	(0.061)	(0.048)	(0.047)	(0.040)
Education	0.027	0.762	0.468	−0.183	−0.054
	(0.387)	(0.884)	(0.548)	(0.531)	(0.463)
Constant	10.790 ***	9.175	5.028	4.927	5.093
	(3.151)	(6.532)	(4.370)	(4.301)	(3.972)
AIC	12004.730	13252.520	10731.480	9687.068	9817.765
Log-Likelihood	−5988.364	−6612.261	−5351.739	−4829.534	−4894.882
Observations	1746	1746	1746	1746	1746

Note: robust standard errors in parentheses; *, **, and *** indicate significance at the 10%, 5%, and 1% significance level, respectively.

## Data Availability

The datasets used and analyzed during the current study are available from the corresponding author on reasonable request.

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
