# Peer review of "Estimates of Household Food Waste by Categories and Their Determinants: Evidence from China"

_foods, 2023, doi:10.3390/foods12040776_

Round 1
Reviewer 1 Report
Food waste generation is one of the key challenge of nowadays. Development of technologies for the utilization of food waste, and reducing the amount of household type food waste has high relevance. But these awareness need detailed and up-to-date date an information related to the type and amount of waste and the influence of socioeconomic factors and consumers behavior, respectively. Therefore, the manuscript has a relevant topic, and MS can provide useful and interesting data and information. Introduction section summarizes well the relevance and the novelty of the study. Section 2 provides a concise but clear literature review related to the research based on relevant references. Online surveys are commonly used for data collection and can provide useful and up-to-date information. As authors mentioned, the design of surveys based on preliminary focus group interviews (including consumers, and professionals, as well). The categorization of food waste is well determined. Logit and Tobit models are appropriate methods for the present research and study. The methods are described well and clearly. The limitations of the study are clearly given (section 6.2). The manuscript contains interesting and valuable data and information.
Comments, suggestions:
It should be given in the MS title that establishments and data derived from households in China.
I recommend the authors to revise the grammar and typing errors in the whole manuscript (see line 34 ’large body’, line 133.sentence strat with RefNo, line 137 missing bracket etc.).
In my opinion presenting of manuscript structure in Introduction in form and with present content is unnecessary (line 81-85).
Establishments in section 5.2 need more detailed discussion with relevant references.
Please discuss briefly the results with international experiences and data, as well.
Reviewer 2 Report
Dear authors,
The aim of this research is to estimate the proportion of whole food and four food categories, and we also consider the whole food and four food types, which may produce behavioural differences under the impact of consumer characteristic factors. Please consider the following commentaries in order to improve the manuscript:
Introduction
- p. 3, lines 120-121: We ask authors to reconsider the fact that “vegetarianism” is a “demographic variable”, as it is a high-level motivation factor.
- p. 3, lines 129-131: Considering the previous research, we ask authors to explicit the trend regarding the differences between urban and rural areas in household food waste.
- p. 3, line 137: Typo error, please close the bracket (in bold), “… associated with age [36, 38]. The education….”
Questionnaire design
- p. 4, lines 151-152: Authors should clarify which general foods are most consumed in China.
- p. 4, lines 152-158: Authors should better describe the questionnaire: for each part of the survey, authors should identify: the number of questions, the nature of the questions (e. g. dichotomic, rating scales…), and the eventual bibliographic reference that was used for the questions elaboration.
- Can authors please identify the “trap questions”?
- For this statement/question: “In general, how much of the following food categories has been discarded in your household in the past week”), did authors clarify to the participants the measure of the variable (units, kilograms, litres)? This is particularly relevant as this question could have different participant interpretations.
- We ask authors to apply the original survey (translated into English) as supplementary material to this manuscript.
p. 4, lines 159-153: Who conducted the focus group? Authors should explain why they had invited different actors (food science experts, food enterprise managers, professionals, food market regulators, and general consumers) to participate in the focus group? Did the interviews transcribed to be analysed? Which framework methodology was used to analyse the interviews (n=42)?
- p. 4, 177-180. The question that report the proportion of household food waste in the past week (“In general, how much of the following food categories has been discarded in your household in the past week”), should be included in the 3.1. Questionnaire design section.
Tobit model
- p. 5, line 198: typo error: separate “is” from “i” We assume that the individual is?,
- p. 5, line 206: authors should confirm this typo error (in bold), as vector X contains all independent variables: “Where vector ?=(?1,…??…,??) contains all dependent variables that could affect…”
Summary statistics
- Authors should clarify if there is representativeness of the sampled population.
Results
- p. 7, lines 250-252: Authors should explain why in terms of the incidence rate of food waste in urban and rural areas, rural areas have a higher incidence rate of the five categories than urban areas?
- p. 9, line 298: Authors should explain why consumers who experience hunger tend to waste food?
Reviewer 3 Report
The manuscript is based on a solid sample and follows a clear structure. The introduction and the literature are clear and straightforward.
The main shortage of the manuscript is that the results are not clearly discussed and interpreted. Among others, one can expect that having experience in starvation results in less food waste, however, the results suggest the opposite - and this is not discussed in the paper. Therefore, a substantial reconsideration of the section of discussion is required.
Moreever, the manuscript has several other shortcomings that should be covered:
1) In the methodological part, there is no information on how the accuracy of respondents' answers were measured (e.g., how did they know what food waste is, how did they measure their proportions etc.).
2) There is no information on the representativeness of the sample.
3) The age intervals of the category of child and elder is not clear
4) It is not clear what the authors mean under staple and snack, as they refer to them as food not consumed by vegetarians (e.g. in line 293)
5) The manuscript refers to times to CCTV news as a validation; however, there is no reference of this source. It might be also considered whether a scientific publication should refer to the public media in this way.
6) A technical note: formatting direct references are incorrect (e.g., " For example, [5] found that..."
Round 2
Reviewer 3 Report
The majority of my concerns were addressed, only two minor remarks remained:
Comment 2: In the methodological part, there is no information on how the accuracy of respondents' answers were measured (e.g., how did they know what food waste is, how did they measure their proportions etc.).
Here I didn't mean the econometric validation of the calculations, but I rather missed information on whether the respondents were informed what to consider food waste (e.g., not to measure the weight of the plastic package, etc.) and whether they were told how to measure the proportion of the waste (e.g., compare the weight of the waste to the raw weight of the unprepared raw materials, etc.)
Comment 6: It is not clear what the authors mean under staple and snack, as they refer to them as food not consumed by vegetarians (e.g. in line 293).
It is still unclear what the authors meant under "staple and snack", maybe some example would be useful to illustrate.
Author Response
Dear reviewer, please see the attachmen.
